# Hedgehog-Related Mutation Causes Bone Malformations with or without Hereditary Gene Mutations

**DOI:** 10.3390/ijms241612903

**Published:** 2023-08-17

**Authors:** Shoko Onodera, Toshifumi Azuma

**Affiliations:** 1Department of Biochemistry, Tokyo Dental College, 2-9-18 Kanda Misaki-cho, Chiyoda-ku, Tokyo 101-0061, Japan; onoderashoko@tdc.ac.jp; 2Oral Health Science Center, Tokyo Dental College, 2-9-18 Kanda Misaki-cho, Chiyoda-ku, Tokyo 101-0061, Japan

**Keywords:** Gorlin syndrome, craniofacial development, hedgehog signaling, gene panel, genetic diagnosis, bone anomaly

## Abstract

The hedgehog (Hh) family consists of numerous signaling mediators that play important roles at various stages of development. Thus, the Hh pathway is essential for bone tissue development and tumorigenesis. Gorlin syndrome is a skeletal and tumorigenic disorder caused by gain-of-function mutations in Hh signaling. In this review, we first present the phenotype of Gorlin syndrome and the relationship between genotype and phenotype in bone and craniofacial tissues, including the causative gene as well as other Hh-related genes. Next, the importance of new diagnostic methods using next-generation sequencing and multiple gene panels will be discussed. We summarize Hh-related genetic disorders, including cilia disease, and the genetics of Hh-related bone diseases.

## 1. Introduction

Bone tissue cells reside in a variety of locations, including perichondrial areas of the postnatal growth plate, the resting zone of the growth plate, and the cranial neural crest (CNC). In addition, mesenchymal stem cells (MSCs) can be found in the bone marrow and periosteum. MSCs capable of differentiating into bone tissue were first identified in the bone marrow, where they were characterized as STRO-1-positive cells by surface markers [1,2]. These MSCs were further classified by surface markers, and then enriched as CD146-positive cells, CD51-positive cells, CD73-positive cells, and PDGFRα-positive cells [3,4]. Bone tissue differentiates from these mesenchymal pluripotent cells in two distinct ways: either through membranous ossification or endochondral ossification, each of which has been analyzed in detail. Particularly in endochondral ossification, a Prrx1-positive mesenchymal stem cell population emerges in a condensed mass of MSCs, which express Sox9 [5]. Chondrocytes, located in the center of the condensed mass of mesenchymal cells, and perichondrocytes, located in the periphery, then develop and differentiate. The central chondrocytes are poorly vascularized and are surrounded by perichondrocytes with rich blood vessels. The growth plate is composed of chondrocytes, and Indian hedgehog (Ihh) secreted by these chondrocytes promotes the growth of hypertrophic chondrocytes. Ihh secreted by chondrocytes stimulates not only hypertrophic chondrocytes, but also perichondrocytes, from which osteoblasts differentiate. This cell population is called osteogenic perichondrocytes. Alternately, mesenchymal stem cells in the bone marrow cavity are found primarily in the vascular-rich epiphyseal and epiphyseal areas and are identified as Gremlin1-positive cells [6,7]. At the boundary with adjacent chondrocyte regions, Gli (Glioma-associated oncogene 1)-positive metaphyseal mesenchymal progenitor cells are present and are highly sensitive to Ihh signaling. Thus, Ihh is an important factor for skeletal stem cells entering the osteoblastic lineage. Gli1, 2, and 3 are known to be orthologous transcription factors of Hedgehog (Hh), and, as noted above, their genetic abnormalities are often associated with skeletal abnormalities, particularly in the head and neck region.

Gorlin syndrome (GS), a well-known genetic and neoplastic disease caused by hedgehog pathway abnormality which is also called nevoid basal cell carcinoma syndrome (NBCCS) or basal cell nevus syndrome (BCNS), is an autosomal-dominant genetic disease caused primarily by *patched1* (*PTCH1*) mutations in the Hh receptor. The syndrome was first described by Gorlin and Goltz in 1960 [8]. This syndrome is characterized by developmental abnormalities such as palmar or plantar pits, bone malformations, medulloblastoma, basal cell carcinoma, and odontogenic keratocysts (OKC). Based on these characteristics, several diagnostic criteria have been established, among which those reported by Kimonis et al. in 2004 are now commonly used [9,10,11,12,13]. These diagnostic criteria consist of major and minor criteria, as shown in Table 1. The major criteria include multiple basal cell carcinomas, early detection of odontogenic keratocysts, palmoplantar pyocystosis, ectopic calcification or thickening of the intrinsic layer, and a family history of GS. Minor criteria include bone abnormalities such as fusion, bifurcation, or rib defects; fibromas of the heart or ovaries; medulloblastoma; lymphocytic cysts; and cleft lip or palate.

Theoretically, two major criteria, or one major and two minor criteria, are sufficient for diagnosis; however, in practice, it is often necessary to confirm the presence of symptoms that meet these criteria in multiple departments. The age and frequency of onset vary greatly, depending on individual symptoms and race. For example, BCC has been reported in 15.2% of Koreans [14], 38% of African Americans, 80% of Caucasians [10], and 76% of Australians [15], and the average age of onset is 20.3 years [15]. The incidence of medulloblastoma in GS patients is very low compared to that in patients with BCC, with an occurrence in up to 5% of GS patients in the first 2 years of life; it generally occurs in early childhood [16]. Shanley et al. reported that OKC were present in 75% of 113 patients with GS, with a mean age at onset of 15.5 years [15]. The most common clinical manifestations in all ages were palmar or plantar pits, present in 87% of patients with GS. There is a relatively high frequency and low age of onset of OKC, palmar pits, and plantar pits in GS; however, in many cases, the symptoms necessary for GS diagnosis are not present at the time of the first symptom’s onset. Therefore, early diagnosis is often difficult. Given the rapid progress in genetic analysis technology in recent years, it is becoming increasingly important to apply genetic analysis for the early diagnosis of GS rather than relying only on clinical symptoms for diagnosis. Advances in the development of sequencing instruments inspired by the Human Genome Project and the corresponding improvements in dry analytical techniques have made genomic information easier, faster, and cheaper to obtain. Diagnoses can now be made without relying solely on clinical symptoms. The early application of genetic analysis for diagnosis may become increasingly important in the future. Furthermore, to gain a deeper understanding of patients with GS, it is necessary to elucidate genetic aspects such as causative and Hh-related genes.

## 2. Genetic Aspects of Gorlin Syndrome

Gorlin syndrome is characterized by genetic abnormalities in negative regulator molecules in the Hh pathway, most frequently PTCH1 [17,18] and, rarely, PTCH2 [19,20] and SUFU [21,22]. These are Hh-pathway-related proteins that coordinate cell–cell communication during development and regeneration and play a crucial role in tumor suppression.

### 2.1. PTCH1

PTCH1 is a human homologue of the patched *Drosophila* segment polarity gene *patched* (PTCH1; MIM 601309), which maps to 9q22.3-q31. The *PTCH1* gene consists of 23 exons encoding 1447 amino acids, which have 12 transmembrane regions, 2 loop domains in the extracellular region, and 1 loop domain in the intracellular region, as shown in Figure 1 [23]. Additionally, the patched transmembrane helices 2–6 are predicted to group together to form a sterol-sensing domain (SSD), which plays a role in cholesterol metabolism and signaling [24].

Ptch1 is a receptor for three types of Hh ligands, namely, Sonic Hh (Shh), Indian Hh, and Desert Hh, and regulates the Hh signaling pathway by repressing its downstream molecule, smoothend (SMO). In the absence of Hh ligands, Ptch1 inhibits SMO translocation through the cytoplasm to the primary cilium. This inhibition induces the phosphorylation of a complex containing a suppressor of fused (Sufu) and full-length Gli; this phosphorylation is performed by several kinases, including protein kinase A (PKA), casein kinase 1 (CK1), and glycogen synthase kinase 3 (GSK3). The proteins of the phosphorylated complex are then either degraded by b-TrCP or targeted to produce truncated repressor forms [25] (Figure 2a). In the presence of Hh ligands, Hh binds to Ptch1, and this binding releases SMO. The released SMO translocates to the primary cilia and accumulates β-Arrestin to activate Gli transcription factors and express target factors, such as *GLI1* and *PTCH1*, which contribute to positive and negative feedback loops. Other target genes involve cell proliferation (*MYC*, *Cyclin D*, and *Cyclin E*) and cell survival (*BCL-2*), as well as several cytokines [26,27] (Figure 2b). This pathway is called the canonical Hh pathway. Mutations in Ptch1 due to hereditary conditions cause an inability to suppress SMO, resulting in constant SMO activation and hyperactivation of the Hh pathway in the absence or presence of Hh (Figure 2c).

This abnormal Hh pathway is thought to induce several phenotypes, such as BCC, OKC, palmar or plantar pits, and ectopic calcification, in the lamellar or falx. *PTCH1* mutations have been detected in approximately 59% to 90% of patients with GS, while the rest of the mutations are found in other genes, such as *SUFU* and *PTCH2* [28,29,30,31,32]. *PTCH1* mutations in GS have been reported in numerous papers since the first publication by Gorlin and Goltz in 1960 [8]. More than 250 *PTCH1* pathogenic mutations have been previously reported without hotspot mutations [33]. Although an association with Hh has been established, and several genetic variants have been reported, their correlation with the pathological phenotype has not been found to date. Therefore, the mechanisms leading to mutation-induced pathogenesis are unknown.

Mutations of *PTCH1* are also implicated in human cancers. The phenotypes of GS include medulloblastoma, multiple nerved basal cell carcinoma, and a benign tumor called a fibroma, which can grow in the heart or in a woman’s ovaries. There is a canonical and a non-canonical Hh pathway in cancer development. Medulloblastoma and nerved basal cell carcinoma develop in a ligand-independent manner, while other colorectal, prostate, liver, and breast cancers develop in a ligand-dependent autocrine/juxtacrine-activated manner [34,35,36]. Whether in GS or not, PTCH1 acts as a tumor suppressor gene. Recently, relationships with immunity and Hh signaling were also reported. Wang et al. reported that PTCH1-mutated tumors promote antitumor immunity through, for example, CD8^+^T cells, activated NK cells, and M1 macrophages in colorectal cancer patents [37]. Analysis of Shh^−/−^ thymus in embryogenesis reveals that SHH is necessary for the efficient proliferation of thymocytes, and for differentiation from the CD4^−^CD8^−^ double-negative thymocytes to the CD4^+^CD8^+^ double-positive stage [38]. The complex signaling makes it difficult to grasp the entire Hh mechanism.

### 2.2. SUFU

*SUFU* genes have been implicated in GS. The human suppressor of the fused gene (SUFU; MIM 607035) consists of 12 exons on chromosome 10q24-25 [39]. *Drosophila* Sufu encodes 468 amino acids with a high-scoring PEST-domain which are rich in proline (P), glutamate (E), serine (S), and threonine (T). The human SUFU protein shares 63% of its sequence with the *Drosophila* Sufu protein and 97% of its sequence with the mouse Sufu protein [40]. SUFU plays a role in the nuclear–cytoplasmic shuttling of Gli transcription factors and is a negative regulator of Hh signaling [41]. Sufu^−/−^ knockout is embryonically lethal like Ptch1 knockout, and Sufu^+/−^ heterozygous mice show a strong similarity to Ptch1^+/−^ heterozygous mice [42]. Sufu^+/−^ heterozygous mice were found to display features of GS, with a distinct skin phenotype and developed 100% penetrance [43]. Subsequently, *SUFU* mutations were found in patients with GS without *PTCH1* mutations [21,22]. Individuals with an *SUFU* pathogenic mutation (33%) have a much higher occurrence of medulloblastoma than those with a *PTCH1* pathogenic variant (2%). An *SUFU* mutation has a much lower occurrence of OKC than a *PTCH1* mutation (62.7%) [44]. According to these results, although both PTCH1 and SUFU function as negative regulators of Hh signaling, these pathogenic genomic mutations implicate different phenotypes in patients with GS. Interestingly, *PTCH1* and *SMO* pathogenic mutations are substantially more frequent in sporadic medulloblastoma than *SUFU* pathogenic mutations are. Recent reports indicate that high expression of the m6A methyltransferase METTL3 is associated with a poor prognosis in MB patients, due to activation of Shh signaling via regulation of the stability and translation of *PTCH1* and *GLI2* RNA [45]. Other research groups have found that the HECT E3 ubiquitin ligase Itch, in conjunction with the adaptor protein β-arrestin2, binds SuFu and promotes its ubiquitylation. This ubiquitylation facilitates the formation of the SuFu/Gli3 complex, which increases the amount of Gli3R and thus maintains inhibition of the Hh pathway [46]. In addition to mouse models, patient-derived iPSCs have also started to be used as disease models. Susantoa et al. reported using induced pluripotent stem cell (iPSC)-derived human neuroepithelial stem cells to construct a medulloblastoma model [47]. In another case, other researchers showed that PTCH1-null-induced pluripotent stem cells exclusively differentiate into immature ectodermal cells with large areas of medulloblastoma-like tissue [48].

These results indicate that genomic *SUFU* mutations derived from GS and somatic *SUFU* mutations derived from sporadic medulloblastoma may have different mechanisms for establishing medulloblastoma.

### 2.3. PTCH2

The *Patched2* gene (PTCH2; MIM 603673) is another rare gene associated with GS. The mouse Ptch2 protein shares 56% homology with mouse Ptch1 [49,50,51]. The human *PTCH2* gene consists of 22 exons encoding 1203 amino acids, which have the same structural domain as *PTCH1* [52]. Both Ptch1 and Ptch2 are functionally redundant. However, the current belief that Ptch2 has limited involvement in Hh signaling is supported by the divergent phenotypes of embryonic lethal Ptch1^−/−^ mice and substantially normal Ptch2^−/−^ mice [42,53]. Patients with GS and *PTCH2* mutations show milder phenotypes than those with *PTCH1* mutations [19,20,54]. Fujii et al. reported that individuals with GS and *PTCH2* mutations do not show typical phenotypes such as palmar/plantar pits, falx calcification, or coarse faces, even when they meet Kimonis’ criteria with multiple OKC and rib anomalies [19]. Casano et al. found variant c.3347C>T (p.Pro1116Leu) in exon 21 of PTCH2 in the proband, and the patient, who did not meet Kimonis’ criteria, had several minor diagnostic features of GS, including macrocephaly, a wide face, and palmar pits [54]. We also reported that several patients with GS have *PTCH2* mutations and *PTCH1* gene mutations [55,56]. Mutations in the PTCH2 gene alone have rarely been reported to be causative genes for GS; however, we found overlapping mutations in the PTCH1 and PTCH2 genes. At present, the correlation between the PTCH1 genotype and phenotype is unknown, and some mutations in PTCH1 may or may not cause GS. There may also be new symptoms due to the duplication of mutations in PTCH1 and PTCH2. Therefore, further investigation is required.

## 3. Hedgehog Function and Its Instability

In addition to the genes responsible, Hh signaling is regulated by several ligands and receptors. Dysfunction of these genes also induces malfunctions in several tissues, especially in osteogenesis and morphogenesis.

### 3.1. DHH, IHH, and SHH

Dhh is involved in male gonadal differentiation and perineural development. In vertebrates, SHH and IHH are the major ligands involved in skeletal development; SHH is primarily involved in the early stages of mesenchymal condensation; and IHH in the progression of endochondral ossification [57,58]. SHH signaling is critical for bone formation, and its dysregulation is responsible for extensive skeletal abnormalities, especially in the limbs, hands, and face. SHH promotes the epithelial–mesenchymal transition of skeletogenic mesenchyme [58]. Shh plays several significant roles in the development of the head processes, notochord, ventrolateral midbrain, and ventral forebrain. In addition, Shh is essential for limb development, including limb budding, anterior–posterior skeleton patterning, and regulation of right-to-left asymmetry [58,59]. SHH also plays an important role in finger patterning and regulates facial development and growth, as well as the differentiation of cranial neural crest cell-derived skeletal structures [60]. The group of diseases caused by SHH gene mutations includes congenital hand deformities; Werner’s syndrome; Acheiropodia; and various forms of polydactyly and syndactyly such as Polydactyly type 1 (PPD1), Polydactyly type 2 (PPD2), and Syndactyly type 4 Haas type [61,62,63]. Laurin–Sandlow syndrome is a severe craniofacial and neurological syndrome, an autosomal-dominant disorder characterized by polysyndactyly of the hands and feet, mirror image duplication of the feet, and nasal defects caused by a heterozygous mutation in an SHH regulatory element (ZRS) that resides in intron 5 of the LMBR1 gene on chromosome 7q36 [64].

Previous studies have reported abnormalities or morphological defects in the craniofacial and long tubular bones, and the progression of several cancers in Shh-deficient mice. Shh-deficient mice lose axial patterning and show an absence of distal limb structures and cyclopia [58]. Shh is expressed in the ventral forebrain neuroepithelium and oral ectoderm, but is absent from the neural crest-derived mesenchyme [65]. Shh signals from the foregut endoderm provide cranial neural crest cells with information on the size, shape, and orientation of the skeletal elements that will eventually form from the pharyngeal arches [66]. Shh is also expressed in the first branchial arch on E9.5 [67,68] and Shh null mice lack significant mandibular development owing to small mesenchymal condensation in Meckel’s cartilage [69]. The loss of Shh in Nkx2.5Cre, Shh^+/−^ mice in the pharyngeal arch resulted in the complete loss of Meckel’s cartilage and tongue at E14.5 and E15.5 [70]. There is a human phenotype similar to that of the Shh mutation. Human holoprosencephaly (HPE), a hereditary disease caused by Shh mutations, presents with a variety of malformations, such as complete absence of the lower jaw and facial cleating [71,72]. The Shh phenomenon is important in cranial facial development.

In contrast, IHH regulates chondrocyte differentiation, promotes endochondral bone growth, and directly stimulates osteoblast precursor cells and the subsequent differentiation of Runx2-positive osteoblasts. It also inhibits GLI3 activity [73,74,75]. IHH promotes hypertrophic chondrocyte differentiation through a negative feedback loop involving IHH-parathyroid hormone-related proteins (PTHrPs). PTHrPs also increase chondrocyte proliferation in the growth plate and osteoblast differentiation later in development [76,77]. Mutations in IHH located in a specific region of the *N*-terminal active fragment cause Brachydactyly A-1 (BDA1), which is characterized by shortness of the middle phalanges of the hands and toes and a shortened stature [78,79,80]. Missense heterozygous mutations in IHH cause type A1 polydactyly, syndactyly with craniosynostosis, syndactyly Lueken type, and acrocapitofemoral dysplasia with reduced IHH signaling in the growth plate along with increased chondrocyte differentiation, resulting in short stature and short limbs [81,82].

The diffusion of Hh ligands across developing limb buds creates a concentration gradient that acts as a morphogenetic factor defining finger development. Homozygous mutations in IHH cause epiphyseal femoral dysplasia, an autosomal-recessive disorder associated with conical bone ends in the hands and hips [82]. In particular, mutations that activate Hh signaling inactivate the GLI3 repressor as well, causing Greig’s head polydactyly syndrome, Pallister–Hall syndrome, and polydactyly of acrosomal polydactyly type 4, as seen in GLI3 mutations [83].

In the cranial facial region, Ihh also governs the development of cranial bases. Ihh knockout mice showed narrow intrasphenoidal and spheno-occipital regions with reduced growth and chondrocyte proliferation in the cranial bases [84]. In another study, Ihh-deficient mice showed reduced bone development and impaired secondary hard-palate ossification, with decreased osteogenic gene expression at E16.5–E17.0 [85]. Similar to that in long bones, Ihh also affects chondrocyte differentiation during facial development. In addition to the endochondral ossification-like limb, IHH affects intramembrane ossification during craniofacial development [86]. In contrast to the Shh mutation and mouse phenotype, IHH mutations in hereditary diseases do not result in a clear facial phenotype. Brachydactyly type 1 is an autosomal-dominant disease caused by heterozygous mutations in IHH [87,88]. The patient presents shortened limbs, but no morphological abnormalities of the facial phenotype. Based on the discrepancy in the phenotypes of mice and humans, we will need more clinical and basic associations in order to understand the function of Ihh in cranial facial development.

### 3.2. KIF3a and IFT

Kinesin family member 3a (Kif3a) is one of the heterotrimeric motor proteins in primary cilia. Once Hh binds to Ptch1, Ptch1 represses SMO, and consequently, SMO moves to the primary cilium. In the primary cilia, there are two conserved, specific microtubule motors: the plus-end-directed heterotrimeric kinesin-2 complex, which consists of KIF3a, KIF3b, and KIF-associated protein 3, is required for anterograde transport from the base to the tip (KAP3) [89]. The minus-end-directed cytoplasmic dynein2 motor, comprising a heavy chain, an intermediate chain, a light intermediate chain, and several light chains, is required for retrograde transport from the tip to the base [90]. Consequently, large and electron-dense IFT trains formed by the protein complexes IFT-A and IFT-B connect these motors to the cargo [91,92]. These actions stimulate SUFU, leading to its targeted expression. The Kiension-2 complex and IFT are important for Hh signaling.

These proteins function as motor and transport proteins of primary cilia. Genetic diseases based on mutations in the genes encoding these molecules are grouped under the disease category of ciliopathy. These are discussed in more detail in a later section.

### 3.3. GLI1, GLI2, and GLI3

GLI1, GLI2, and GLI3 are transcription factors that belong to the Gli family and are required for the transduction of the Hh pathway in mammals [93]. Gli1 is an Hh target gene that acts as a transcriptional activator, whereas Gli2 and Gli3 act as both activators and repressors [94]. Gli1 is not considered to be required for mouse development, as *Gli1*^−/−^ mutants are alive, survive from birth to adulthood, and have a normal phenotype [95,96]. However, heterozygotic deficient Gli1 mice show decreased bone mass with reduced bone formation and accelerated bone resorption, suggesting the uncoupling of bone metabolism [97]. In humans, mutations of the Gli1 gene have been reported in some genetic disorders, including polydactyly of the biphalangeal thumb and/or hallux, postaxial polydactyly type A, postaxial polydactyly type B, and Ellis–van Creveld syndrome. The variants of Gli2, include Culler-Jones syndrome, which leads to abnormal development of the brain structures, limbs, midline face, cleft lip, and partial palate [98]. In both human and mouse models, aberrations in Gli3 result in craniofacial dysmorphisms. Greig cephalopolysyndactly syndrome, which results in metopic synostosis and is marked by polydactyly and hypertelorism, is one effect of altered Gli3 sequencing [99,100,101]. Pallister–Hall syndrome, another phenotype caused by a Gli3 mutation, is characterized by disrupted midline development and craniofacial abnormalities, including a short nose with a flat nasal bridge and cleft palate [102,103]. Genetic diseases, such as acrocallosal syndrome with craniofacial abnormalities and tibial hemimelia, are also known to occur. The causative genes of acrocallosal syndrome are not only *GLI3,* but also *KIF7,* which is another member of the kinesin family. KIF7 deficiency causes improper GLI3 processing with a diminished Gli3 repressive form (GLI3R), leading to inappropriate activation of SHH target genes in both humans and animals [104,105]. Furthermore, Putoux et al. showed that Kif7^−/−^ Gli3Δ699 mice exclusively produced the repressive isoform of Gli3 (GLI3R) rescued by increasing GLI3R activity, indicating that decreased GLI3R signaling is responsible for the ACLS features in these mice [106]. These results suggest that Hh signaling is important for osteogenesis.

## 4. Hedgehog Signaling and Ciliopathy

All vertebrate tissues and cell types can produce primary cilia, also called sensory cilia, inside the cell; they receive stimuli from outside the cell and transmit signals. A variety of signals can be received by specific ciliary receptors to sense physical stimuli (e.g., mechanical stress), light, hormones, chemokines, growth factors (e.g., somatostatin, stromal cell-derived factor 1 [SDF-1], platelet-derived growth factor (PDGF)), or regulators of signaling pathways such as Shh and Wnts. The genes responsible for ciliopathies are highly conserved, and their encoded proteins interact dynamically in the cilia, basal bodies, centrosomes, and mitotic spindles. Cilia are widespread and present in almost all tissues, and mutations in these genetic alterations affect a variety of tissues and organ systems [107].

The relationship between Hh signaling and cilia was revealed in a mutagenesis screening analysis of mice that identified mutations in the IFT gene as the cause of the Hh mutant phenotype [108,109]. Mutant mice of *Nphp7*/*Glis2*, which encodes the transcription factor Gli-similar protein 2 (GLIS2), also show severe renal atrophy and fibrosis similar to human renal atrophy 73, highlighting the link between ciliopathy and Hh [110].

Some ciliopathies, such as Ellis–van Creveld syndrome, Jeune asphyxiating thoracic dystrophy, and short rib polydactyly syndrome, are characterized by severe skeletal and craniofacial dysplasia, with short bones, a narrow chest with short ribs, and polydactyly [111,112]. Ciliopathies with craniofacial defects include Bardet–Biedl syndrome (BBS), oro-facial-digital syndrome (OFD1), Meckel or Meckel–Gruber syndrome (MKS), Joubert syndrome, and Ellis–van Creveld syndrome [113]. BBS is an autosomal-recessive disease mainly characterized by retinal dystrophy, obesity, post-axial polydactyly, renal dysfunction, learning difficulties, and hypogonadism [114]. It also showed subtle craniofacial dysmorphisms and oral/dental animalities like crowding, hypodontia, and a high arched palate with a presence of over 50% [115,116,117]. OFD1 (OMIM #311200) is an X-linked inherited disease by CXORF5 characterized by the malformation of the face, oral cavity, hands, and feet [118]. Because primary cilia are involved in osteoblast alignment and polarization, as well as osteoblast differentiation and bone formation, a link between ciliary dysfunction and skeletal and craniofacial bone defects was assumed. Ciliary proteins (e.g., IFT80, IFT88, KIF3A, EVC) have also been reported to play substantial roles in osteoblast differentiation and function [119,120]. Defects in IFT80 in osteoblast progenitor cells cause significant growth retardation and osteopenia associated with impaired osteoblast differentiation; IFT80 deficiency blocks cilia-dependent canonical Hh-Gli signaling and overactivates the cilia-independent non-canonical Hh-Gαi-RhoA pathway, which inhibits osteoblast differentiation. In addition to the regulation of osteoblast differentiation, ciliary proteins also regulate cell polarity and alignment. A recent study showed that deletion of IFT20 in osteoprogenitors results in the disruption of osteoblast polarity, which inhibits osteoblast formation, thus causing disruption of collagen fiber formation and resulting in reduced bone strength and stiffness.

### 4.1. Primary Cilia as Mechanosensor

Primary cilia function as important mechanosensors in osteoblasts by regulating multiple pathways [120]; when cilia are lost due to the deletion of IFT88, osteogenic responses to mechanical stimuli are lost as well [121]. In contrast, mechanical stimulation promotes the migration of bone marrow cells to the bone surface and their differentiation into osteoblast lineages. Disruption of cilia by the deletion of KIF3A in cells of the cell lineage inhibits mechanical loading-induced bone formation [122]. Osteocalcin-conditioned Kif3a-null (Kif3a^OC-cKO^) mice develop osteopenia with impaired bone mineral density, trabecular bone volume, and cortical thickness due to impaired osteoblast function [123]. Other conditional Kif3a-deficient mice display cranial base growth retardation and dysmorphogenesis at the neonatal stage. These mice lack Kif3a in the cartilage and show unusual growth plates without the typical zones of chondrocyte proliferation [124]. Wnt1 conditional Kif3a-null mice have stunted lower jaws, and 33% of them lack primary and secondary palates in embryos [125]. One of the IFT-B subunits, IFT88, has also been reported to be a critical factor for cranial facial development. Ift88-deficient mice present defects in neural tube patterning, craniofacial abnormalities, polydactyly, and left–right axis determination [126,127]. Deletion of the ciliary protein Ift88 in the mesenchyme results in ectopic mandibular bone formation [128]. In patients with non-syndromic cleft lip, two IFT88 iatrogenic variants, rs9509311 and rs2497490, are associated with development [129]. In contrast to IFT-B proteins, the role of IFT-A proteins in osteoblasts is less well defined. However, gene mutation in IFT-A molecules disrupts Hh signaling, i.e., the IFT-A gene that often exhibits enhanced Hh signaling. For example, hypo-morphic missense mutant mice of the IFT144 gene from *N*-ethyl-*N*-nitrosourea show impaired ciliogenesis, impaired limb development, impaired somatic patterning of ribs, and facial and cleft palate defects. Mutations in IFT144 show ligand-independent expansion of Hh signaling and defects in Shh/GREM1/FGF interactions, especially in the limbs and facial ridges [130]. IFT140, one of the IFT-A components, has been found to be important for endochondral bone development [131]. Lineage-specific deletion of IFT140 in osteoblast precursors causes marked growth retardation and osteopenia with a dwarfism phenotype and promotes age-related bone loss [132].

### 4.2. Primary Cilia and Chondrocyte

Since some osteoblasts develop from endochondral stem cells following chondrogenesis, primary cilia and ciliary proteins in the cartilage play an important role in osteogenesis. Kitami et al. recently reported that IFT20 (one of the components of IFT-B) is important for maintaining the condylar cartilage [133]. Loss of IFT20 in chondrocytes significantly attenuated HH signaling through the cilia and significantly reduced the level of X-type collagen production. IFT46 is another component of the IFT-B complex; Park I et al. suggest that IFT46 regulates ciliogenesis and craniofacial development by modulating the Wnt/PCP and Shh signaling pathways [134]. KIF3A, KIF5B, and KIF7 have been reported to play important roles in cartilage development. The loss of KIF5B in chondrocytes results in disorganized growth plates, as well as delayed and defective cytoplasmic division, which impairs proliferation and differentiation [135]. Primary cilia play an essential role in the development and function of intervertebral disc cartilage. Mice lacking IFT80 in chondrocytes exhibit intervertebral disc degeneration.

### 4.3. Primary Cilia in Craniofacial Development

Ciliopathies are known to show craniofacial abnormalities such as a cleft lip/palate, hypertrichosis/hypochondriasis, micrognathism, and craniosynostosis [111,136]. During the development of the palate, maxilla, and mandible, loss of IFT88 in cranial neural crest (CNC) lineage cells disrupts ciliogenesis and reduces the proliferation of neural crest cells on the palatal shelf during palatogenesis; this results in craniofacial defects and death at birth [137]. In addition, IFT88 deficiency in CNC-derived mesenchymal cells leads to downregulation of Shh signaling during palatogenesis and upregulation of Wnt signaling, leading to cleft palate defects, as well as ectopic hyperostosis of the maxillary process and abnormal apoptosis [138]. More recently, Kitamura A et al. suggested that one important aspect of defects in craniofacial development is the disruption of Hh signaling [128]. Defects in IFT20 disrupt PDGF signaling, resulting in decreased osteoblast proliferation and bone mineralization. They also cause severe craniofacial malformations, such as hypertelorism, abnormal enlargement of the facial midline, mandibular and maxillary hypoplasia, and lack of a palatal shelf and tongue [139]. In addition, IFT20 deficiency in CNC-derived mesenchymal cells increases FKBP65 chaperone levels and inhibits the biosynthesis of collagen, resulting in fragile bones [140].

### 4.4. Primary Cilia in Tooth Development

The primary cilium has been shown through ACⅢ and acetylated tubulin immunohistochemical staining at various stages of tooth formation. In fetal enamel organs, the primary cilium is expressed in enamel knots, which are signaling centers that signal mesenchymal tissues [117]. In other tooth-related tissues, the hypofunctional periodontal ligament (PDL) downregulates the expression of *BBS7* by delaying cell migration and suppressing cell angiogenesis [141]. Recent studies have shown that the primary cilium has a close relationship not only with Hh, but also with WNT [142], fibroblast growth factor (FGF) [143], and PDGF [144]. Their functional correlation in the morphogenesis and differentiation of several tissues may have a significant impact.

During tooth formation, dental pulp stem cells (DPSCs) differentiate into odontoblasts, which produce dentin. Since primary cilia are present in the tooth embryo and oral tissues, mutations in ciliary proteins often cause ciliopathies with tooth dysplasia [145]. Primary cilia in DPSCs elongate during tooth formation by regulating fibroblast growth factor 2 (FGF2)-FGF receptor 1 (FGFR1) signaling. Deletion of IFT80 in DPSCs disrupts ciliogenesis and inhibits odontogenic differentiation by disrupting FGF receptor expression and FGF signaling, Hh signaling, and their cross-talk [146]. In addition, primary cilia promote the polarization of odontoblasts during odontogenesis, which is an important step in the formation of dentin tubular structures. Furthermore, in mice lacking IFT80 in odontogenic cells, DPSC proliferation is reduced, molar development is impaired, molar roots are shortened, and incisor eruption is delayed. IFT140 is also highly expressed during odontoblast differentiation, and has been shown to play an important role in tooth development. Loss of IFT140 in odontoblasts also disrupts primary ciliogenesis and Shh signaling, resulting in impaired odontoblast differentiation and dentin deposition [147]. Thus, these findings indicate that FGF and Hh signaling are important for the regulation of primary cilia in tooth development.

## 5. Genetic Analysis

Gene panel systems, which can detect specific gene regions using next-generation sequencing (NGS), have been used for cancer prediction and diagnosis. Similar to cancer gene panels, comprehensive panels have been used for the diagnosis of GS. Morita et al. established a custom Haloplex panel containing genes involved in the Hh-related pathways PTCH1, PTCH2, SHH, SUFU, SMO, GLI1, GLI2, and GLI3 in the GS [148]. Nakamura Y et al. constructed a more compact genetic panel which comprised PTCH1, PTCH2, SUFU, and SMO for NGS [56]. Using this panel, we confirmed that one of the blood relatives who did not appear to have a diagnostic phenotype had the same *PTCH1* mutation as that found in a patient with GS. Interestingly, both panels included *SMO*, another receptor for the Hh ligand that functions as a positive regulator. SMO mutations are more common in patients with sporadic BCC [149,150,151]. The phenotype and X-ray criteria of Kimonis’ diagnosis are still powerful tools in practice; the following scenarios are suggested for genetic testing: confirmation of diagnosis in patients lacking sufficient clinical diagnostic criteria, such as patients with PTCH2 mutations; predictive testing for at-risk patients with an affected family member, but who do not meet the clinical criteria; and prenatal or early-childhood testing in the presence of a known familial mutation. This genomic diagnosis will be of great value for early diagnosis, even if individuals do not show a GS phenotype.

## 6. Conclusions

The Hh signaling pathway is critical for the initiation of bone tissue development and is required for many types of tissue development. It is also a key factor in the development of BCC and medulloblastoma. Thus, the Hh pathway plays a major role in the pathogenesis of GS. However, owing to the complexity of the onset mechanism in relation to the phenotype and mutation, numerous issues need to be resolved. This review addresses several of these issues. In recent years, many studies have reported using iPS, cells unique to GS, to develop modern therapeutics. Both mechanism and mutation analyses are required in future research, and these results are highly promising.

## Figures and Tables

**Figure 1 ijms-24-12903-f001:**
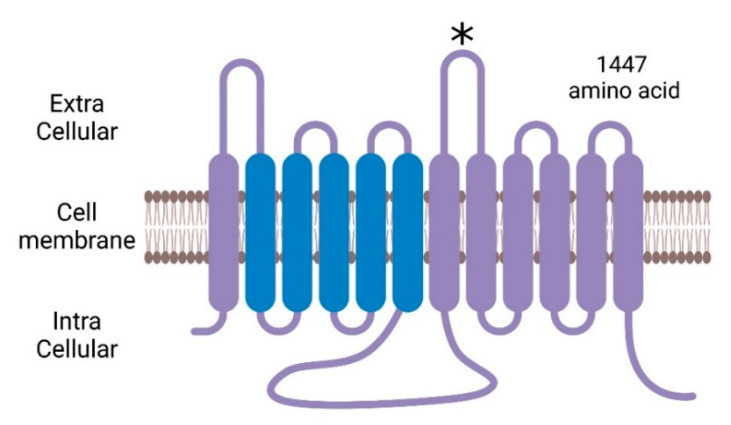
The structure of Patched1 (PTCH1). There are 2 extracellular loops and 1 intracellular loop. Blue parts indicate the sterol sensing domain (SSD). The asterisk on the 2nd loop is the binding part of the Hh ligand. The figure was created using BioRender.com (accessed on 1 August 2023).

**Figure 2 ijms-24-12903-f002:**
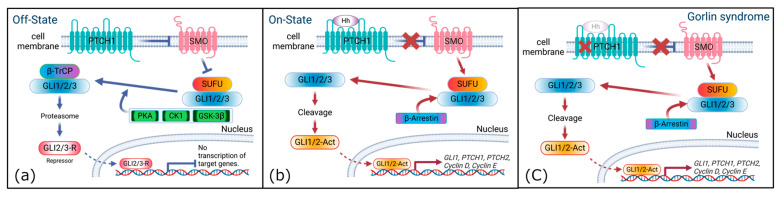
Hedgehog (Hh) signaling pathway, normal and with Gorlin syndrome. Off-state panel (**a**) shows the absence of the Hh ligand. On-state panel (**b**) shows the presence of the Hh ligand. Gorlin syndrome (**c**) shows the condition in pathogenic signaling. Mutations in Ptch1 cause an inability to suppress SMO, resulting in constant SMO activation and Hh pathway hyperactivation with the absence or presence of the Hh ligand. The figure was created using BioRender.com (accessed on 1 August 2023).

**Table 1 ijms-24-12903-t001:** Gorlin syndrome (GS) diagnostic criteria (from Kimonis V.E. et al. [10]).

**Major Criteria**
1. More than two BCCs or one under the age of 20 years.
2. Odontogenic keratocysts of the jaw that are proven by histology.
3. Three or more palmar or plantar pits.
4. Bilamellar calcification of the falx cerebri.
5. Bifid, fused, or markedly splayed ribs.
6. First-degree relative with Gorlin syndrome.
**Minor Criteria**
1. Macrocephaly determined after adjustment for height.
2. Congenital malformations: cleft lip or palate, frontal bossing, “coarse face”, moderate or severe hypertelorism.
3. Other skeletal abnormalities: Sprengel deformity, marked pectus deformity, marked syndactyly of the digits.
4. Radiological abnormalities: bridging of the sella turcica, vertebral anomalies such as hemivertebrae, fusion or elongation of the vertebral bodies, modeling defects of the hands and feet, or flame-shaped lucencies of the hands or feet.
5. Ovarian fibroma.
6. Medulloblastoma.

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
