# Peer review of "Hedgehog-Related Mutation Causes Bone Malformations with or without Hereditary Gene Mutations"

_ijms, 2023, doi:10.3390/ijms241612903_

Round 1

Reviewer 1 Report

The authors have put together a comprehensive review on hedgehog signaling pathway and implications of its dysregulation. The review is well written and structured appropriately. There are some minor errors that need to be addressed. 

1.      Line 42, 118-119 and 376 needs language correction, line 312 is written in present tense, needs to be corrected to past tense.

2.      PDGF was abbreviated in line 447 however it was previously used in line 340, should be abbreviated there.

3.      Line 173 – is the percentage 0 ?

4.      Please add reference for lines 476-479. Check line 479 for error.

5.      For figure 2, the authors have used right, middle and left in reference to the subsections, which is okay, but it might be better to follow the traditional denotations by using a,b and c for the subsections in the figure.

6.      The authors mention the modern therapeutics that can be used for treatment in the conclusion section. I feel if the authors included another subsection dedicated to this, it would immensely improve the impact of the article.

Some minor correction required as noted above.

Author Response

Dear reviewer,

On behalf of all authors, we thank the reviewer for providing thoughtful comments.  Attached please find a word file with our responses to specific comments.

We look forward your response.

Best regards,

Shoko Onodera

Reviewer 2 Report

1) Please include a bar chart indicating the frequency of hedgehog mutations in Gorlin syndrome 

2) I suggest the authors add the hedgehog pathway and refer to genes described in the text for easy interpretation 

Author Response

Dear reviewer,

On behalf of all authors, we thank the reviewer for providing thoughtful comments. Our responses to specific comments are shown below.

1) Please include a bar chart indicating the frequency of hedgehog mutations in Gorlin syndrome 

Response: Thank you for your advice. We concur with your viewpoint. Although, it is hard to make a bar chart indicating the frequency of hedgehog mutation in Gorlin syndrome. Because Gorlin syndrome is known that there is no relationship between phenotype and mutations. In the manuscript, we presented percentage of each phenotype and each mutation from previous studies, but we believe this is insufficient to summarize the entire tendency of mutation and phenotype distribution of Gorlin syndrome. In addition, due to the rarity of Gorlin syndrome, the number of each case is not enough for discussion. From these reasons, it is hard to make exact bar chart of Gorlin syndrome. I applicate your idea for understanding the concept of Gorlin syndrome.

2) I suggest the authors add the hedgehog pathway and refer to genes described in the text for easy interpretation 

Thank you for your comment. The explanation of Hh pathway is written in Line 125 – 141.

We look forward to your response.

Best regards,

Shoko Onodera

Reviewer 3 Report

Dear Authors,

I would like to congratulate you on this developmental review concerning genetic mutations and the genotype-phenotype correlation.
Please consider the minor suggestions within the paper I am sending you back.

Best regards,

Dr. Andrea Gama

Minor revision suggested.

Author Response

Dear Reviewer,

On behalf of all authors, we thank the reviewer for providing thoughtful comments and advice. The corrections are shown below and heighted in the manuscript.

  • Line 54 and 140: We corrected Golts to Goltz.
  • In Major criteria no.6 of Table 1: We corrected GS syndrome to Gorlin syndrome
  • Line 159: We corrected Golrin to Gorlin.
  • Line 251: We corrected “Dhh, IHH, and Shh” to “DHH, IHH, and SHH”.
  • Line 323: We corrected “Kif3a and IFT” to “KIF3a and IFT”.
  • Line 340: We corrected “Gli1, Gli2, and Gli3” to “GLI1, GLI2, and GLI3”.
  • Line 344: We corrected “Gli1−/−” to “Gli1−/−”.
  • Line 381: We corrected “NPHP7/GLIS2” to “Nphp7/Glis2”.

We look forward to your response.

Best regards.

Shoko Onodera